# Structural Analysis and Antioxidant and Immunoregulatory Activities of an Exopolysaccharide Isolated from *Bifidobacterium longum* subsp. *longum* XZ01

**DOI:** 10.3390/molecules28217448

**Published:** 2023-11-06

**Authors:** Xingyuan Zhang, Jing Gong, Wenyi Huang, Wen Liu, Chong Ma, Rongyao Liang, Ye Chen, Zhiyong Xie, Pei Li, Qiongfeng Liao

**Affiliations:** 1School of Pharmaceutical Sciences, Guangzhou University of Chinese Medicine, Guangzhou 510006, China; xyzhang9906@163.com (X.Z.); gongjing@gzucm.edu.cn (J.G.); 13825357502@163.com (W.H.); liangrongyao114@163.com (R.L.); chenye_0928@163.com (Y.C.); 2School of Pharmaceutical Sciences (Shenzhen), Sun Yat-sen University, Shenzhen 518106, China; liuw257@mail2.sysu.edu.cn (W.L.); mach53@mail2.sysu.edu.cn (C.M.); xiezhy@mail.sysu.edu.cn (Z.X.)

**Keywords:** *Bifidobacterium longum* subsp. *longum* XZ01, exopolysaccharide, isolation, structural analysis, antioxidant activity, immunoregulatory activity

## Abstract

*Bifidobacterium longum* subsp. *longum* XZ01 (BLSL1) is a new strain (isolated from the intestines of healthy people and deposited with the preservation number GDMCC 61618). An exopolysaccharide, S-EPS-1, was successfully isolated from the strain and then systematically investigated for the first time. Some structural features of S-EPS-1 were analyzed by chemical component, HPLC, ultraviolet, infrared, and nuclear magnetic resonance spectrum analyses. These analyses revealed that S-EPS-1 is a neutral heteropolysaccharide with an α-configuration. It contains mainly mannose and glucose, as well as small amounts of rhamnose and galactose. The molecular weight of S-EPS-1 was calculated to be 638 kDa. Several immunoregulatory activity assays indicated that S-EPS-1 could increase proliferation, phagocytosis, and NO production in vitro. In addition, S-EPS-1 could upregulate the expression of cytokines at the mRNA level through TLR4-mediated activation of the NF-κB signaling pathway in RAW 264.7 cells. Finally, S-EPS-1 was demonstrated to exhibit antioxidant activity by ABTS^+•^ scavenging, DPPH^•^ scavenging, and ferric-ion reducing power assays. Furthermore, S-EPS-1 can protect cells from oxidative stress and shows no cytotoxicity. These beneficial effects can be partly attributed to its antioxidant ability. Thus, the antioxidant S-EPS-1 may be applied as a functional food in the future.

## 1. Introduction

Polysaccharides, as biological macromolecules, can be derived from plants, animals, and microbes. In particular, microbial polysaccharides have been reported to exhibit a wide range of biological activities, including antioxidant, immunoregulatory, and antitumor effects [1]. Microbial exopolysaccharides (EPSs), particularly those derived from probiotics, have attracted increasing attention due to their immunostimulant potential and minimal side effects [2,3]. Immune cell functions, however, are well known to be closely linked to the generation of reactive oxygen species (ROS), which may trigger cellular oxidative stress [4,5,6,7]. Therefore, a sufficient supply of neutralizing antioxidants can protect immune cells from oxidative stress and maintain their optimal function [8,9,10]. Hence, the immunoregulatory effects of EPSs may be associated with their antioxidant activities.

Bifidobacterium, the dominant bacterium in the human intestinal tract, not only plays a beneficial role in host health but also has a close relationship with the structural composition and function of the intestinal microbiota. Studies have confirmed that bifidobacterium possesses various physiological activities, such as activities that alleviate intestinal inflammation, promote tumor immunity, and ameliorate diabetes [11]. Despite its numerous probiotic effects, Bifidobacterium is a living microorganism and its effects on the host in vivo cannot be accurately predicted and controlled due to variations in the intestinal environment. Complications associated with bifidobacterium have been reported, including bacteremia, sepsis, pneumonia, pericarditis, and urinary tract infections [12]. Moreover, multidrug-resistant probiotics have emerged due to the inappropriate use of antibiotics in clinical practice, posing a significant issue that must be urgently addressed. Some Bifidobacterium strains have shown obvious resistance to antibiotics [13], highlighting the risks associated with the use of live probiotics. Additionally, studies have indicated that the material basis for bifidobacterium to interact with the host and exhibit physiological activities may be derived from the EPSs produced by bifidobacterium [14]. Consequently, research efforts have begun to focus on the active metabolites of bifidobacteria. Bifidobacterium EPSs refers to the macromolecular polymers secreted by bifidobacteria into their growth environment [15]. These EPSs have garnered extensive attention due to their excellent characteristics and diverse biological activities, such as their lack of toxicity, ability to prevent oxidation, immune regulation activity, and ability to metabolize intestinal microorganisms [14,16]. To date, most studies have concentrated on EPSs derived from lactic acid bacteria (LAB) and bifidobacterium, considering their generally recognized safe status. However, compared to LAB EPSs, research on bifidobacterium EPSs remains relatively scarce, limiting the development of bifidobacterium EPSs to a large extent.

In this study, a new bacterial strain with EPS-producing characteristics was isolated in our laboratory and was further identified as Bifidobacterium longum subsp. longum XZ01. The EPSs were extracted and purified into different fractions. Some structural features of one purified fraction, designated S-EPS-1, were analyzed through UV, FT-IR, HPLC, high-performance gel permeation chromatography, and NMR. Furthermore, the immunoregulatory and antioxidant activities of S-EPS-1 were evaluated in vitro. The results obtained from this study could contribute to the comprehensive utilization of S-EPS-1 and promote the use of exopolysaccharides as natural antioxidants, which show potential for application in the food industry.

## 2. Results and Discussion

### 2.1. Identification of Bacterial Strain XZ01

The Gram staining results (Appendix A) indicated that the bacterial strain XZ01 was Gram-positive and possesses a rod-shaped or bifurcated micromorphology, which is a typical characteristic of bifidobacterium. Meanwhile, the bacterial strain XZ01 showed its growth pH in the range of 3–9. BLAST was used to compare the 16S rRNA sequences of strain XZ01 with other bacteria in the NCBI database. The comparison sequences with high similarity were selected and a phylogenetic tree was drawn by MEGA X. According to the phylogenetic tree (Appendix A), strain XZ01 and *Bifidobacterium longum* subsp. *longum* JCM1217 were located in the same branch and have a close genetic relationship. Therefore, strain XZ01 was identified as *B. longum* subsp. *longum* XZ01. And the bacterial strain XZ01 has been deposited in the Guangdong Microbial Culture Collection Center (GDMCC) under accession number GDMCC 61618.

### 2.2. Isolation and Purification of S-EPS-1

Crude polysaccharides (C-EPSs) were obtained through a series of steps, including precipitation, protein removal, and decolorization. Subsequently, C-EPSs were eluted to generate the following primary components: EPS-1, EPS-2, and EPS-3 (Figure 1a). Among them, EPS-1 exhibited the highest purity and was further purified using a Sephacryl S-300HR column (Figure 1b), yielding purified S-EPS-1. Due to the higher purity, S-EPS-1 was selected for subsequent structural and activity studies.

### 2.3. Purity and Molecular Weight Determination of S-EPS-1

After the UV spectrum of S-EPS-1 (Figure 1c) was analyzed within the wavelength range of 190–400 nm, a minimal absorption was observed at 260 nm, 280 nm, and 320 nm. This indicates that negligible amounts of protein, nucleic acid, and pigment were present, suggesting that S-EPS-1 was effectively purified. Additionally, the total sugar and protein contents of S-EPS-1 were determined using the phenol sulfuric acid method and the BCA protein quantitative method, respectively. The results revealed that S-EPS-1 had a total sugar content of 99.20 ± 1.21% and a low protein content, aligning with the UV analysis findings.

To determine the molecular weight of S-EPS-1, HPGPC was employed. As depicted in Figure 1d, S-EPS-1 exhibited a single peak, indicating the higher purity of S-EPS-1. The molecular weight of S-EPS-1 was determined to be 6.38 × 10^5^ Da based on the standard curve. Notably, the molecular weight of bifidobacterium EPSs typically falls within the range of 10^4^ to 1.5 × 10^6^ Da [17,18,19,20], and S-EPS-1’s molecular weight falls within this range, thus confirming its identity.

### 2.4. Structural Features

#### 2.4.1. Monosaccharide Composition of S-EPS-1

The composition of monosaccharides in S-EPS-1 was analyzed using HPLC. As shown in Figure 2a, mannose and glucose were the predominant sugar components in S-EPS-1. The relative proportions of monosaccharides in S-EPS-1 were calculated to be 11.85:0.46:5.60:0.68 (mannose, rhamnose, glucose, and galactose). Additionally, no uronic acid was detected in S-EPS-1. This composition confirms that S-EPS-1 is a neutral heteropolysaccharide. Notably, heteropolysaccharide production is common among bifidobacterial species [20,21]. Corresponding with our findings, EPSs synthesized by *Bifidobacterium animalis* RH were also found to contain glucose, galactose, rhamnose, and mannose in a ratio of 1:4:2:3 [22,23].

#### 2.4.2. FT-IR Spectrum of S-EPS-1

FT-IR analysis was conducted to determine the functional groups and glycoside bond configurations present in S-EPS-1. The FT-IR spectra (Figure 2b) exhibited broad and strong absorption bands at approximately 3296 cm^−1^, assigned to O-H stretching vibrations, and 2932 cm^−1^, assigned to C-H stretching vibrations [24]. These absorption peaks are characteristic of polysaccharides. The strong absorption peak observed at 1645 cm^−1^ was attributed to water absorption [25], indicating the presence of bound water in S-EPS-1. Furthermore, the absorption peak at 1375 cm^−1^ was associated with C-O stretching vibrations and C-H bending vibrations in S-EPS-1 [26].

Moreover, the prominent absorption peaks at 1023 and 912 cm^−1^ indicated the presence of pyranose in S-EPS-1 [27]. The absorption peak at 809 cm^−1^ suggested the presence of mannose pyranose or galactose pyranose [28], aligning with the analysis of monosaccharide composition. Importantly, there was no absorption observed at 890 cm^−1^, indicating the absence of a β configuration in the monosaccharides present in S-EPS-1 [29]. These results further suggested that S-EPS-1 was an α-configuration polysaccharide containing mannose and glucose residues.

#### 2.4.3. NMR Analysis of S-EPS-1

NMR spectroscopy was employed as a powerful tool to elucidate the structural features of S-EPS-1. The anomeric proton region in the NMR spectrum (Figure 3a) displayed seven distinct proton signals at 5.33, 5.22, 5.07, 5.04, 4.97, 4.89, and 4.83 ppm, which were labeled as A, B, C, D, E, F, and G, respectively. Notably, no signal was observed at δ 5.40 ppm, indicating that all the sugar residues in S-EPS-1 were in pyranose form [30,31], consistent with the findings from FT-IR spectroscopy. Furthermore, the HSQC spectrum indicated that the anomeric carbon signals at 99.54 and 100.54 ppm were respectively correlated with the anomeric proton signals δ 5.33 ppm (A) and δ 5.22 ppm (B), while the anomeric carbon signals at δ 102.15 ppm were correlated with the proton signals at δ 5.07 ppm (C) and δ 4.97 ppm (E) (Figure 4). However, in the anomeric carbon signal region (Figure 3b), six signals were evident, with chemical shifts at δ 99.54, 100.54, 102.15, 98.18, 98.56, and 99.34 ppm. These shifts indicated that S-EPS-1 primarily consisted of α-configurations [30,32,33], which aligned with the ^1^H NMR results. The α-configurations can be easily digested by human beings, in contrast to herbivores (e.g., cows), because of the presence of α-glucosidase in human cells [34].

#### 2.4.4. SEM Analysis

SEM images of S-EPS-1 revealed a surface that appeared rough and uneven, characterized by numerous small pores (Appendix A). Furthermore, the presence of loosely distributed small fragments in the images suggested the favorable water solubility of S-EPS-1 [35]. The water solubility, along with the aforementioned human digestive potential, actually promised that S-EPS-1 could be developed as a new functional food, from the angle of chemistry.

### 2.5. Immunoregulatory Activities of S-EPS-1

#### 2.5.1. Effect on Cytokine Expressions

When macrophages are activated, they can produce various immune factors to enhance immunomodulatory effects. IL-1β, IL-6, and TNF-α are crucial cytokines expressed by activated macrophages that regulate the immune response in the host body [36]. We investigated the transcriptional levels of IL-1β, IL-6, and TNF-α in RAW 264.7 cells stimulated by S-EPS-1. As depicted in Figure 5a–c, S-EPS-1 dose-dependently increased the mRNA expression of IL-1β, IL-6, and TNF-α in RAW 264.7 cells compared to the blank control group. This indicates that S-EPS-1 can enhance the expression of these cytokines, thereby exhibiting immunostimulatory effects. 

#### 2.5.2. Effect on Nuclear Translocation of p65

NF-κB, a classical transcription factor, plays a crucial role in the activation of macrophages, as it regulates the expression of genes involved in various cellular processes, including cell growth, apoptotic immunity, and the inflammatory response [37]. One prominent feature of NF-κB activation is the translocation of NF-κB from the cytoplasm into the nucleus. We employed confocal laser microscopy to observe the nuclear translocation of p65 in S-EPS-1-stimulated RAW 264.7 cells. As depicted in Figure 6, in the control group, the green fluorescent-labeled p65 protein was located outside of the RAW 264.7 nucleus, and there was minimal overlap between the green fluorescence and the blue fluorescent-labeled nucleus. This suggests that p65 protein was predominantly present in the cytoplasm and that the NF-κB signaling pathway was not activated. In contrast, both the S-EPS-1- and lipopolysaccharide (LPS)-treated groups exhibited significant overlap of green and blue fluorescence, indicating that p65 protein was activated and translocated from the cytoplasm to the nucleus in these groups. This finding suggests that S-EPS-1 may be involved in the activation of the NF-κB signaling pathway.

#### 2.5.3. TLR4/NF-κB Signaling Pathway

Due to its high molecular weight, EPSs are unable to pass through the cell membrane. Therefore, the effect of EPSs on RAW 264.7 cells is likely recognized by pattern recognition receptors on the cell surface. Toll-like receptor 4 (TLR4), known for its involvement in the regulation of immunity and inflammation, also plays a role in the activation of the downstream NF-κB signaling pathway [38]. TLR4 has been proved to be a receptor that recognizes most polysaccharides. As shown in Figure 7a, phosphorylation of the p65 protein in RAW 264.7 cells increased upon stimulation with S-EPS-1, indicating the activation of the NF-κB signaling pathway. Similarly, Figure 7b demonstrates that the expression level of TLR4 protein also increased in a concentration-dependent manner after S-EPS-1 stimulation, suggesting the activation of the TLR4 signaling pathway. These results suggest that S-EPS-1 may activate macrophage RAW 264.7 cells by regulating the TLR4/NF-κB pathway, thereby exerting its immunomodulatory activity. Collectively, these findings indicate that S-EPS-1 possesses antioxidant properties and can modulate immunity by acting on all processes of the TLR4/NF-κB pathway [39,40,41].

#### 2.5.4. Effect on Phagocytic Activity

Phagocytosis is a crucial and essential process for macrophages to defend against and eliminate pathogens. To investigate the effect of S-EPS-1 on the phagocytic activity of RAW 264.7 cells, a neutral red assay was conducted at different concentrations. As depicted in Figure 8b, the control group exhibited primarily round-shaped RAW 264.7 cells with minimal neutral red uptake. In contrast, the cells treated with S-EPS-1 or LPS showed noticeable enlargement and evident neutral red uptake, indicating that S-EPS-1 enhanced the phagocytic activity of RAW 264.7 cells. The quantitative results presented in Figure 8c further confirmed that S-EPS-1 could enhance the phagocytosis of RAW 264.7 cells for neutral red within the concentration range of 25–200 μg/mL. Moreover, an increase in concentration resulted in stronger phagocytic activity of RAW 264.7 cells, demonstrating a concentration-dependent effect. These findings suggest that S-EPS-1 has the potential to improve the phagocytic activity of RAW 264.7 cells, thereby activating their immune response.

### 2.6. Cytoprotective Effects of S-EPS-1 on Cell Viability

#### 2.6.1. Cell Viability

As illustrated in Figure 8a, S-EPS-1 exhibited no toxic effect on RAW 264.7 cells after 24 h treatment in a concentration range of 25–200 μg/mL and could promote the proliferation of RAW 264.7 cells in a concentration-dependent manner, which exhibited the same tendency as LPS.

#### 2.6.2. NO Production Regulation

NO is a vital signaling molecule released by activated macrophages and plays a role in regulating immune responses in vivo. Hence, we measured the NO content in the supernatant of RAW 264.7 cells after treatment with S-EPS-1. As illustrated in Figure 8d, S-EPS-1 stimulation significantly increased the production of NO in RAW 264.7 cells compared to the control group. However, even at the highest concentration, the NO production stimulated by S-EPS-1 did not surpass that of LPS. This suggests that S-EPS-1 exerted a mild effect on RAW 264.7 cells and could prevent excessive NO-induced cell apoptosis. Based on these findings, it can be concluded that S-EPS-1 exhibited an immunomodulatory effect, which can be attributed to its antioxidant or cytoprotective activity [42,43,44,45,46].

### 2.7. Antioxidant Activity of S-EPS-1

As shown in Figure 9a, the ferric reducing power of S-EPS-1 exhibited a significant increase with increasing concentrations, reaching 49.71% at 150 μg/mL. Notably, this value was approximately half that of Trolox at the same concentration, indicating the potent reducing capacity of S-EPS-1. As seen in Figure 9b,c, the radical scavenging rates of S-EPS-1 increased in a concentration-dependent manner in ABTS^+•^ and DPPH^•^ assays. The maximal scavenging rates of S-EPS-1 for 2,2′-azinobis (3-ethylbenzothiazoline-6-sulfonic acid) free radicals (ABTS^+•^) and 1,1-Diphenyl-2-picrylhydrazyl free radicals (DPPH^•^) were 81.52% and 63.79%, respectively. The IC_50_ values (Table 1) of DPPH^•^ radical scavenging activities for S-EPS-1, melatonin, and Trolox were 740.2, 1068.1, and 2.4 μg/mL, respectively. Moreover, the DPPH^•^ radical scavenging rate of S-EPS-1 was found to be stronger than that of melatonin, indicating its superior antioxidant effect. These results suggest that S-EPS-1 exhibits a higher antioxidant effect. When considering mechanistic chemistry, ABTS^+•^ scavenging, DPPH^•^ scavenging, and ferric-ion reducing antioxidant power (FRAP) are mediated through electron transfer (ET) and hydrogen atom transfer (HAT, or hydrogen-donating) mechanisms [47,48,49,50]. Considering the high antioxidant potential demonstrated by S-EPS-1, we believe that its immunomodulatory properties are partly attributable to this antioxidant activity [51,52,53].

## 3. Materials and Methods

### 3.1. Materials and Reagents

RAW264.7 macrophages were purchased from the American Type Culture Collection (ATCC). LPS and monosaccharide standards were purchased from Sigma-Aldrich (Saint Louis, MO, USA). Dextran standards were purchased from Aladdin Biochemical Technology Co., Ltd. (Shanghai, China). The DPPH and ABTS were obtained from Yuanye Technology Co., Ltd. (Shanghai, China). All remaining chemical reagents were provided by Beijing Solarbio Science and Technology Co., Ltd. (Beijing, China) and were analytically pure.

### 3.2. Culture Condition and Identification of Bacterial Strain XZ01

The bacterial strain XZ01 was isolated and maintained in the laboratory. It was revived on an MRS plate at 37 °C for 48 h under anaerobic conditions. Following activation, a single pure colony was transferred to MRS liquid medium for culture expansion under the same conditions for subsequent experiments. To identify the bacterial strain XZ01, Gram staining, physiological and biochemical characterization, and 16S rRNA gene sequencing were performed. The generated sequences were then compared for sequence homology using the BLAST function in the NCBI database to determine the species of the bacterial strain XZ01. Additionally, phylogenetic analysis and the construction of a phylogenetic tree were conducted using the neighbor-joining method in Molecular Evolutionary Genetics Analysis (MEGA X, Version 10.2.4) software.

### 3.3. Isolation and Purification of S-EPS-1

EPSs were extracted and isolated according to a previous study [54]. Briefly, bacteria were cultured in MRS liquid medium with a 2% inoculum and allowed to grow for two generations. The bacterial cells were then removed by centrifugation, and the culture supernatant was collected and heat-treated to inactivate enzymes. The supernatant was concentrated to 1/10 of the original volume, and three times the volume of pre-chilled ethanol was added to precipitate the EPSs. The precipitated fraction was collected and dissolved in an appropriate amount of ultrapure water to obtain a crude extract. The crude extract was then subjected to deproteinization using Sevag reagent (chloroform–butanol = 4:1) and decolorization using AB-8 macroporous absorption resin. The resulting solution was dialyzed and freeze-dried to obtain crude EPSs, referred to as C-EPSs.

C-EPSs were loaded on DEAE Cellulose DE-52 and eluted with 0, 0.05, 0.1, 0.3, and 0.5 M NaCl solutions at a flow rate of 1 mL/min, successively. Each 8 mL fraction was automatically collected, and the presence of EPSs in the eluent was confirmed using the phenol sulfuric acid method. The EPS fractions eluted at the same peak were combined, dialyzed, and freeze-dried. The fraction eluted with ultrapure water (EPS-1) was further purified using a Sephacryl S-300HR column and eluted with 0.1 M NaCl solution (4 mL/tube) at a flow rate of 0.5 mL/min to obtain a purer fraction, designated S-EPS-1.

### 3.4. Chemical Properties and Determination of Molecular Weight

The total sugar content of S-EPS-1 was determined by the phenol sulfuric acid method with glucose as the reference [55]. The protein content of S-EPS-1 was determined according to the operation of the BCA protein quantitative kit, and bovine serum albumin (BSA) was applied as the standard.

The molecular weight and purity of S-EPS-1 were determined by HPGPC. S-EPS-1 was eluted with ultrapure water at a flow rate of 1 mL/min through a PolySep-GFC-P 4000 (300 × 7.8 mm, Phenomenex, Torrance, CA, USA) column and monitored by an evaporative light-scattering detector at 35 °C. Dextran standards with different molecular weights were utilized for the calibration of molecular weight.

### 3.5. Structural Analysis

#### 3.5.1. Monosaccharide Composition Analysis

The monosaccharide composition of S-EPS-1 was determined by PMP pre-column derivatization by a previous method with minor modifications [56]. In brief, S-EPS-1 was subjected to hydrolysis using trifluoroacetic acid (TFA) and subsequently derivatized with PMP. Finally, the solution was analyzed via an HPLC-UV instrument with a C_18_ column (4.6 × 250 mm, 5 μm, Waters, Milford, MA, USA) using 0.05 M phosphate buffer (pH 6.7) and acetonitrile (83:17, *v*/*v*) as the mobile phase.

#### 3.5.2. UV-Vis, FT-IR, and NMR Spectra Analysis

S-EPS-1 solution (1 mg/mL) was scanned from 190 to 400 nm on a UV spectrophotometer (UV1280, Shimadzu, Kawasaki, Japan). To obtain the FT-IR spectrum of S-EPS-1, an FT-IR spectrometer (Perkin Elmer, Waltham, MA, USA) was employed. The spectrum was recorded in the range of 4000 cm^−1^ to 400 cm^−1^. 

For NMR analysis, 30 mg of S-EPS-1 was subjected to three rounds of exchange with deuterium oxide (D_2_O) using repeated freezing and thawing cycles. Subsequently, it was dissolved in 0.55 mL of D_2_O. The ^1^H NMR, ^13^C NMR, and HSQC spectra were recorded using a 600 MHz NMR spectrometer (AVANCE Ⅲ, Bruker, Zürich, Switzerland).

#### 3.5.3. SEM Analysis

The morphology of S-EPS-1 was examined using a field emission scanning electron microscope (Merlin, Forchtenberg, Germany). Prior to imaging, the S-EPS-1 samples were sputter-coated with gold to enhance conductivity and were then observed under a high-vacuum condition with an accelerating voltage of 5 kV. Images were captured at various magnifications, including 50×, 100×, 1000×, and 3000×, providing detailed visualizations of the surface structure and features of S-EPS-1.

### 3.6. Immunoregulatory Activities of S-EPS-1

#### 3.6.1. Cell Culture and Treatment

RAW264.7 macrophages were grown in DMEM coupled with 10% Fetal Bovine Serum (FBS) and 1% streptomycin/penicillin in an atmosphere of 5% CO_2_ at 37 °C. Cells were allowed to be cultured overnight to adherence. LPS (1 μg/mL) was applied as a positive control, and different concentrations of S-EPS-1 (25–200 μg/mL) were used as the experimental groups. Cells treated with complete DMEM medium were considered as blank controls.

#### 3.6.2. Determination of Cytokine Expression

The cytokine expressions at the gene level of macrophages were evaluated by real-time fluorescence quantitative PCR. Briefly, cells were incubated in a 6-well plate at a density of 5 × 10^5^ cells/mL (2 mL/well) and treated with DMEM medium, S-EPS-1 (50, 100, and 200 μg/mL), and LPS (1 μg/mL) for 24 h. The total RNA was extracted using TRIzol reagent and changed to cDNA by reverse transcription. The cDNA was then amplified on a fluorescent quantitative PCR instrument (LightCycler 96, Roche, Basel, Switzerland) using the SYBR Real-time PCR kit. The conditions of qPCR were as follows: initial denaturation at 95 °C for 30 s, followed by 40 cycles of 95 °C for 5 s, 60 °C for 10 s, and 75 °C for 15 s. The primers used in the study are listed in Appendix A. The mRNA expressions of cytokines were standardized to GAPDH. The results are expressed according to the 2^−ΔΔCt^ method.

#### 3.6.3. Immunofluorescence Assay

After being stimulated with LPS (1 μg/mL), complete DMEM medium, and S-EPS-1 (100 μg/mL) for 6 h, cells were treated with 4% paraformaldehyde for 15 min and permeabilized with 0.25% Triton X-100 for 10 min. Next, the cells were treated with 1% BSA for 1 h, followed by the addition of antibody p65 overnight at 4 °C. After that, the cells were incubated with fluorescent secondary antibody for 60 min. Finally, 2-(4-amidinophenyl)-6-indolecarbamidine dihydrochloride (DAPI) was used to stain the cell nuclei and the cells were observed with a confocal microscope (LSM880, Zeiss, Oberkochen, Germany).

#### 3.6.4. Determination of Phagocytic Activity

The phagocytic function of macrophages was assessed using the neutral red uptake assay. RAW264.7 macrophages (100 μL) were seeded in a 96-well plate at a density of 1 × 10^5^ cells per well. The cells were then stimulated with LPS (1 μg/mL), S-EPS-1 (25–200 μg/mL), and complete DMEM medium for 24 h. After incubation, the supernatants were discarded, and 100 μL of a 0.075% neutral red solution was added to each well in the dark. The plate was incubated for 45 min at 37 °C. Subsequently, the cells were observed under a microscope (Nikon Eclipse Ts2, Tokyo, Japan). Afterward, the cells were washed three times with PBS, and 100 μL of cell lysates (ethanol: acetic acid = 1:1, *v*/*v*) was added to each well and incubated for 2 h at room temperature. Finally, the absorbance of the solution was measured at 540 nm.

### 3.7. Cytoprotective Effects of S-EPS-1 

#### 3.7.1. Cytotoxicity Assay

In short, cells were incubated in a 96-well plate (1 × 10^5^ cells/well, 100 μL/well) overnight, then treated with complete DMEM medium, S-EPS-1 (25, 50, 100, and 200 μg/mL), and LPS (1 μg/mL) for 24 h. Then, 20 μL methylthiazolyldiphenyl-tetrazolium bromide (MTT) solution (5 mg/mL) was added to these wells, and incubation continued for 4 h. After incubation, the cell supernatant was removed, and 200 μL of DMSO was added to dissolve the formazan crystals by shaking for 10 min. The absorbance of the resulting solution was measured at 490 nm.

#### 3.7.2. Determination of NO Production

Griess reagent was used to determine the production of NO. In brief, cells were seeded in 96-well plate at a density of 2 × 10^4^ cells/well (100 μL/well), and after being incubated overnight, the cells were treated as described in Section 3.6.1. The subsequent steps were performed according to the instructions provided in the kit. Finally, the optical density was recorded at 540 nm.

### 3.8. Antioxidant Activity

#### 3.8.1. Ferric-Ion Reducing Antioxidant Power

The ferric cyanide (Fe^3+^) reducing power of S-EPS-1 was determined following the method described by Benzie and Strain [57]. To prepare the FRAP reagent, 300 mM acetate, pH 3.6; glacial acetic acid buffer; 20 mM ferric chloride (FeCl_3_•6H_2_O); and 10 mM 4,6-tripryridyls-triazine (TPTZ) were prepared. These three solutions were combined in the ratio 10:1:1 (*v*/*v*/*v*). For the FRAP assay, 30 µL of the sample was mixed with 264 µL of the FRAP reagent and incubated at 37 °C for 30 min. The absorbance was then measured at 593 nm. Trolox and melatonin were used as positive controls. 

The relative reducing ability of the sample was calculated using the following formula:$$relative\;reducing\;effect\%=\frac{A-A_{min}}{A_{max}-A_{min}}\times 100\%$$
where *A_max_* represents the maximum absorbance and *A_min_* represents the minimum absorbance in the test. A represents the absorbance of the sample.

Based on the obtained data, dose–response curves were constructed to determine the IC_50_ values (in μg/mL) (Table 1).

#### 3.8.2. Scavenging Ability of S-EPS-1 on ABTS^+•^

The scavenging activities of S-EPS-1 on ABTS^+•^ were assessed using a modified version of a previously established method [58]. To measure the scavenging activity, 165 µL of the ABTS^+•^ solution was mixed with 40 µL of the sample solution and incubated at room temperature for a specified time. The absorbance at 734 nm was then measured. Trolox and melatonin served as the positive controls. The percentage inhibition of the samples was calculated using the following formula:$$Inhibition\%=\frac{A_{0}-A}{A_{0}}\times 100\%$$
where A_0_ represents the absorbance at 734 nm without any samples and A represents the absorbance at 734 nm in the presence of the samples.

#### 3.8.3. Scavenging Ability of S-EPS-1 on DPPH^•^ Radicals

The DPPH^•^ radical scavenging activity was measured as described in [59], with some modifications. Briefly, 150 µL DPPH^•^ solution (0.1 M) was mixed with 100 µL various concentrations of samples dissolved in 95% ethanol. The mixture was kept at room temperature for 30 min and then measured with a spectrophotometer (Spectra Max i3X, Canberra, Australia) at 517 nm. Trolox and melatonin served as the positive controls. The DPPH^•^ inhibition percentage of the samples was calculated:$$Inhibition\%=\frac{A_{0}-A}{A_{0}}\times 100\%$$
where A_0_ is the absorbance without samples, while A is the absorbance with samples.

### 3.9. Statistical Analysis

The IC_50_ values were calculated using linear regression analysis. In the experiment, the data were expressed in the form of means ± SDs and processed statistically by one-way ANOVA using IBM SPSS 23.0. A *p*-value less than 0.05 was considered statistically significant. 

## 4. Conclusions

From *B. longum* subsp. *longum* XZ01, a novel EPS-producing strain, S-EPS-1, has been successfully isolated for the first time. S-EPS-1 is a neutral and water-soluble heteropolysaccharide with a molecular weight of 6.38 × 10^5^ Da and consists of mannose, glucose, and other sugar residues. These sugars are constructed via α-configuration linkage. Moreover, S-EPS-1 can play an important role in immune regulation via antioxidant mechanisms. This study not only provides a new idea for the development of new strains of probiotics, but also provides a solid basis for the commercial transformation of probiotic EPSs in the functional food industry. 

## Figures and Tables

**Figure 1 molecules-28-07448-f001:**
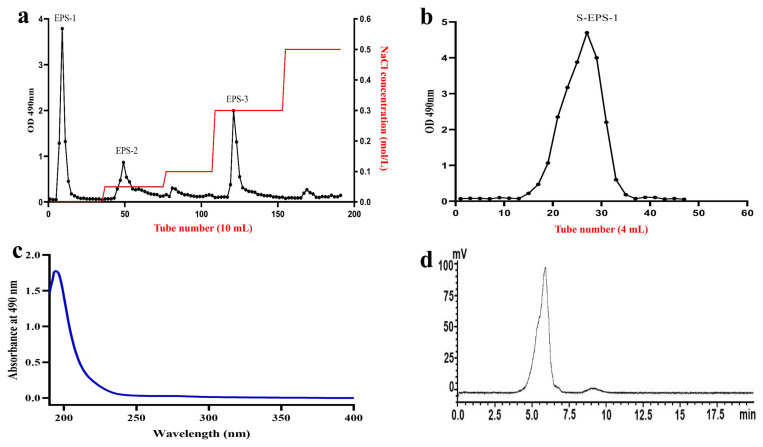
Gradient elution curve of C-EPSs by DEAE-Cellulose 52 (**a**), elution curve of EPS-1 by Sephacryl S-300 HR (**b**), ultraviolet spectrum of S-EPS-1 (**c**), and HPGPC chromatogram of S-EPS-1 (**d**).

**Figure 2 molecules-28-07448-f002:**
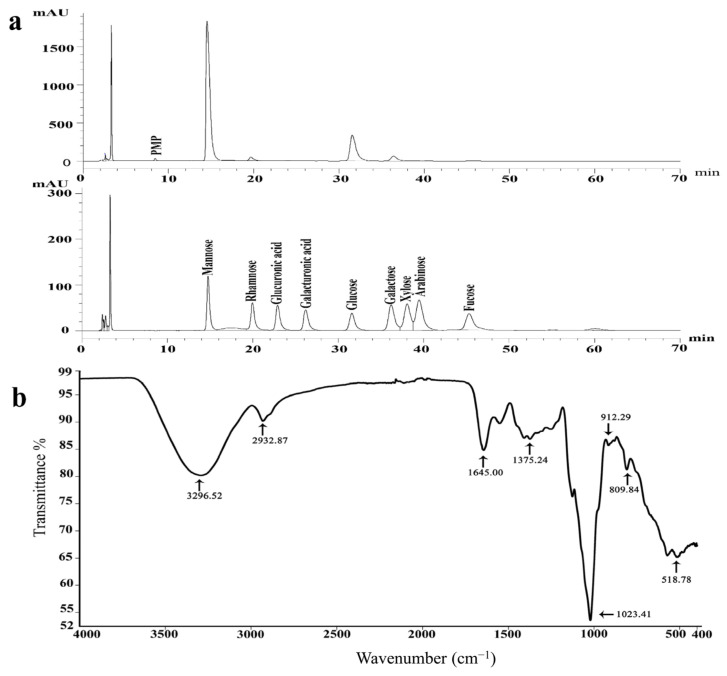
HPLC chromatogram of 1−phenyl−3−methyl−5−pyrazolone (PMP) derivatives of mixed monosaccharide standards and HPLC chromatogram of PMP derivatives of S−EPS−1 (peaks: 1, D−Mannose; 2, L−Rhamnose; 3, D−Glucuronic acid; 4, D−Galacturonic acid; 5, D−Glucose; 6, D−Galactose; 7, L−Xylose; 8, L−Arabinose; 9: D−Fucose) (**a**) and FT−IR spectrum of S−EPS−1 (**b**).

**Figure 3 molecules-28-07448-f003:**
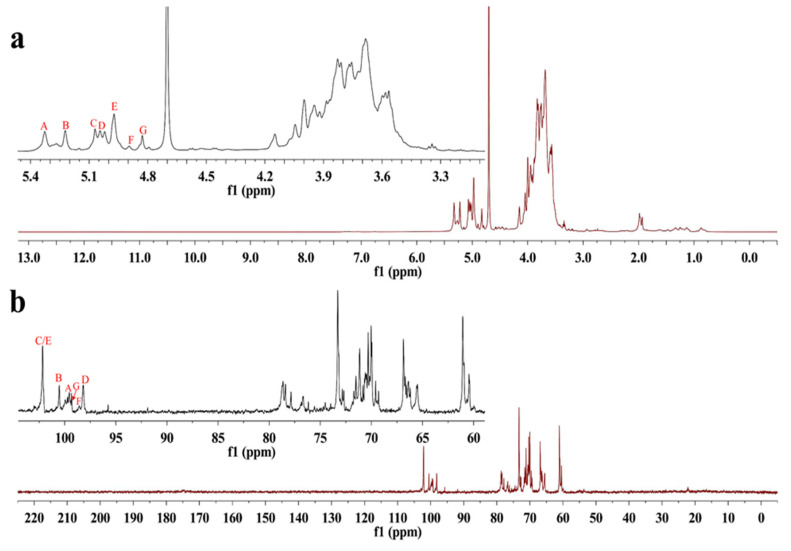
^1^H NMR spectra (**a**) and ^13^C NMR spectra (**b**) of S-EPS-1.

**Figure 4 molecules-28-07448-f004:**
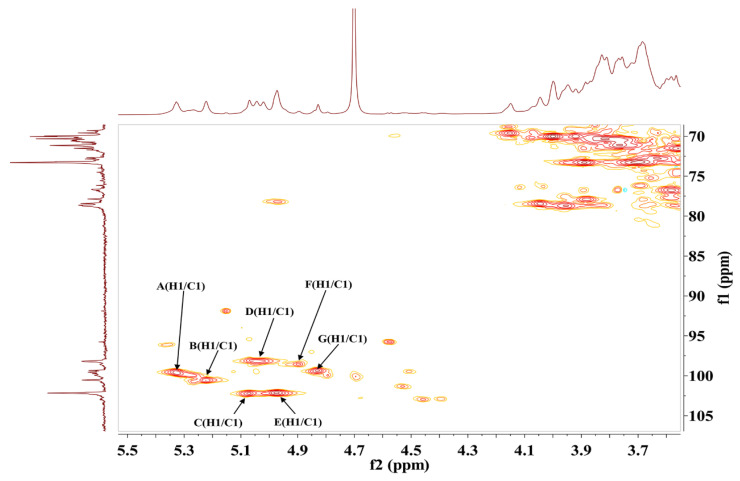
HSQC spectra of S-EPS-1.

**Figure 5 molecules-28-07448-f005:**
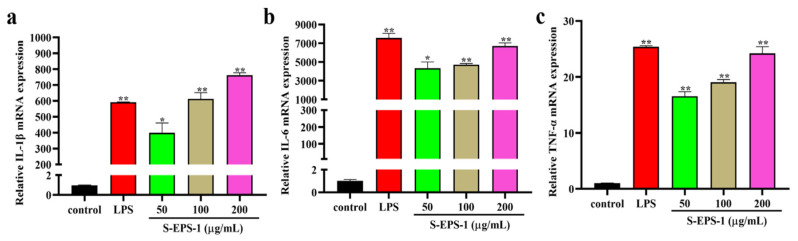
Effects of S-EPS-1 on the expression of cytokines in RAW 264.7 cells. IL-1β (**a**), IL-6 (**b**), and TNF-α (**c**). Compared with control group, * *p* < 0.05, ** *p* < 0.01.

**Figure 6 molecules-28-07448-f006:**
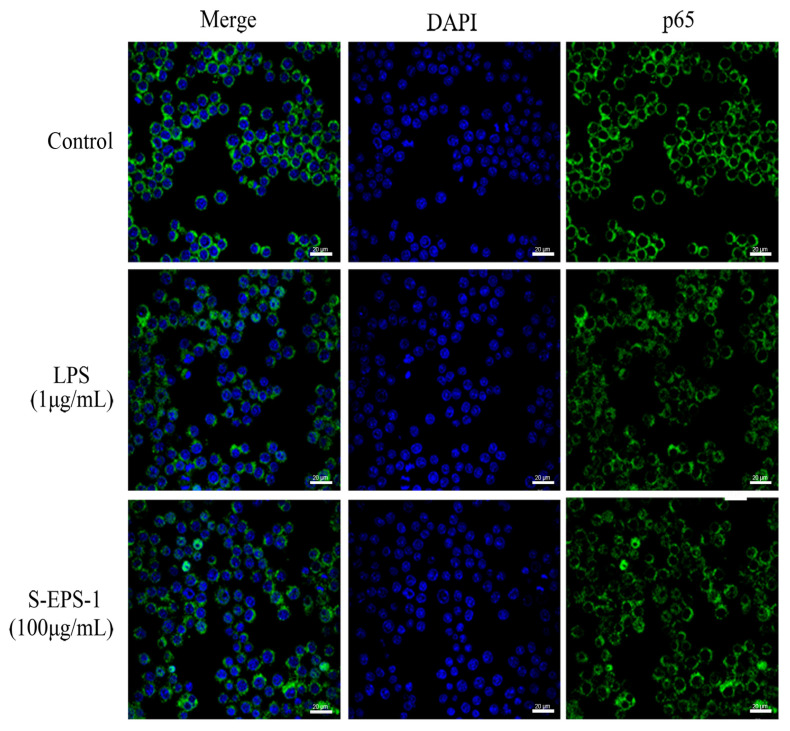
Effects of S-EPS-1 on nuclear translocation of NF-κB in RAW 264.7 cells. The red triangles in the figures indicate nuclear translocation of NF-κB.

**Figure 7 molecules-28-07448-f007:**
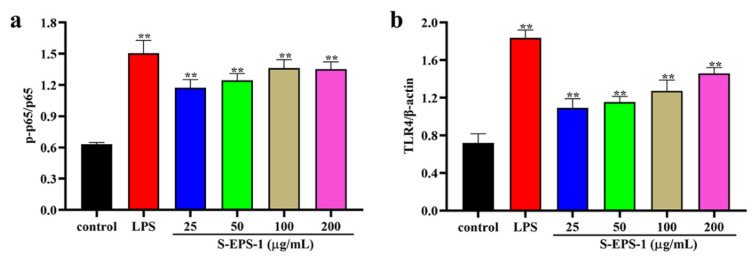
Expression ratio of p-p65 to NF-κB p65 (**a**) and expression of TLR4 (**b**). Compared with control group, ** *p* < 0.01.

**Figure 8 molecules-28-07448-f008:**
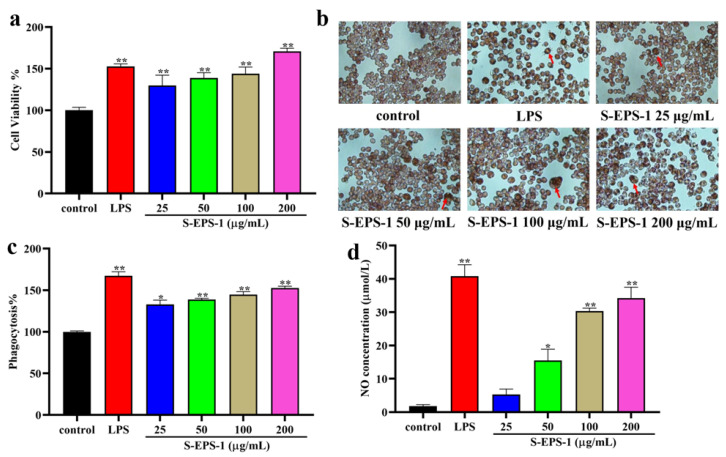
Effects of S-EPS-1 on the cell viability (**a**), phagocytosis of neutral red (**b**), phagocytic activity (**c**), and NO production (**d**) in RAW 264.7 cells. The red arrows indicate the phagocytosis of neutral red in RAW 264.7 cells. Compared with control group, * *p* < 0.05, ** *p* < 0.01.

**Figure 9 molecules-28-07448-f009:**
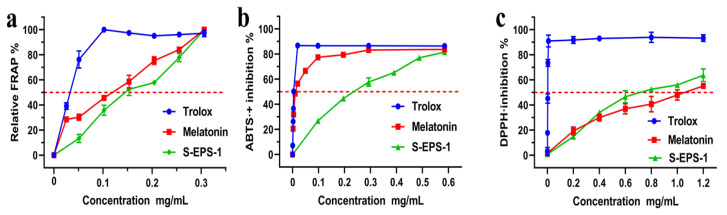
The dose–response curves of S-EPS-1 in a series of antioxidant assays: (**a**) FRAP (ferric-ion reducing antioxidant power), (**b**) ABTS^•+^ scavenging assay, and (**c**) DPPH^•^ scavenging assay. Each value is expressed as the mean ± SD (*n* = 3); Trolox and melatonin were used as the positive controls.

**Table 1 molecules-28-07448-t001:** The IC_50_ values of S-EPS-1 and the positive controls.

Assays	S-EPS-1 (μg/mL)	Positive Controls	
		Trolox (μg/mL)	Melatonin (μg/mL)
ABTS^+•^ scavenging	221.8 ± 2.5 ^b^	4.7 ± 0.1 ^b^	14.1 ± 0.4 ^b^
DPPH^•^ scavenging	740.2 ± 14.9 ^b^	2.4 ± 0.1 ^b^	1068.1 ± 171.3 ^b^
FRAP	141.3 ± 3.8 ^a^	30.7 ± 2.1 ^a^	89.5 ± 4.9 ^a^

The IC_50_ value is defined as the concentration of 50% effect percentage and expressed as a mean ± SD (*n* = 3). Mean values with different superscripts (^a^, ^b^) in the same row are significantly different (*p* < 0.05), while those with the same superscripts are not significantly different (*p* > 0.05).

## Data Availability

Data are contained within the article.

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
