# Peer review of "Structural Analysis and Antioxidant and Immunoregulatory Activities of an Exopolysaccharide Isolated from Bifidobacterium longum subsp. longum XZ01"

_molecules, 2023, doi:10.3390/molecules28217448_

Round 1

Reviewer 1 Report

Comments and Suggestions for Authors

Zhang et al. reported the chemical components of Bifidobacterium longum subsp. longum XZ01 using various analytical techniques including HPLC, ultraviolet, infrared, and nuclear magnetic resonance spectrum analyses. The results revealed that it primarily consisted of mannose and glucose, with small amounts of rhamnose and galactose.

Moreover, the immunoregulatory activity assays demonstrated that S-EPS-1 exhibited the ability to enhance proliferation, phagocytosis, and NO production. Furthermore, it was observed to upregulate the expression of cytokines at the mRNA level through TLR4-mediated activation of the NF-κB signaling pathway in RAW 264.7 cells.

The study exhibits good experimental design and provides an intriguing perspective on the new Bifidobacterium longum subsp. longum XZ01 strain.

Therefore, I suggest making a minor revision.

Comments on the Quality of English Language

Extensive editing of English language required

Reviewer 2 Report

Comments and Suggestions for Authors

Dear Authors,
Congratulations on the article and the results obtained. The demonstration of immunoregulatory activity of the polysaccharide obtained on the Raw 264.7 cell line could be considered as an inflammatory stimulus like other polysaccharides of the bacterial wall, such as LPS, since the activation of NF-kb mediated by binding to the TRL receptor was detected -4?

Reviewer 3 Report

Comments and Suggestions for Authors

The manuscript entitled “Structural Characteristics, Antioxidant, and Immunoregulatory Activities of an exopolysaccharide isolated from Bifidobacterium longum subsp. longum XZ01” contains an interesting piece of research work that, in my opinion, is well worth publishing. A complete set of experiments has been applied to characterize a polysaccharide of novel origin. These experiments are well described and the results presented in a suitable way. The use of English is generally good. And the concussions are supported by the presented data and could be of general utility to other workers.

   My (few) concerns are mostly about figure legends and their arrangement, and about the not clearly disclosed origin of the bacteria employed.

   I enclose a pdf file with my comments and corrections. Deletions are in red and additions and suggestions, in blue. Please, disregard the alterations in the manuscript, due to the passage from pdf to Word, and from Word to pdf, to introduce my comments.

In summary, I consider that this article can be published with minor corrections.

Reviewer 4 Report

Comments and Suggestions for Authors

Manuscript number: molecules-2511878

The authors isolated an exopolysaccharide from Bifidobacterium longum subsp. longum strain XZ01, characterized some structural elements, and determined antioxidant and immunoregulatory activities.

The manuscript is relatively well written, but the structure determination work is not convincing. Bacterial polysaccharides usually have well-defined repeating units composed of a few residues. It is questioned whether the isolated EPS really comes from the bacterium, given its high mannose content and the fact that production was performed in a growth medium rich in yeast extract.

Specific comments:

ll. 36-38

Sentence incomplete

ll. 93, 109-110, 333

A single peak on Sephacryl does not imply that the EPS is pure. The NMR spectra showed many minor peaks (impurities) which were not assigned.

Figure 1

Delete “The ” from axis titles.

ll. 103, 329-330, 335

“sulfuric acid” instead of “sulfate”

ll. 107-108

Abbreviation defined twice: HPGPC

ll.119-120

Repetition of previous sentence

ll. 127, 212, 278

Abbreviations not defined at first occurrence but later: PMP, LPS, ABTS, DPPH

l. 154

“anomeric” instead of “terminal”

l. 161

“COSY (Figure 5)”

ll. 166-168, 174-176

It is not possible to make these assignments from the COSY spectrum as displayed in Figure 5. The region between δ ca. 3.5 to 4.1 is too crowded near the diagonal to extract useful information. The same is true for the other residues. TOCSY is normally acquired for this purpose.

l. 170, Table 1

δ 62.53 is impossible for C-2 of glucose. The same is true for δ 61.59 for C-2 of galactose.

ll. 175, 176

Position 5 of Manp and Galp (pyranoses) is not available for a glycosidic linkage.

Table 1

What is the chemical shift reference?

Very often, the two protons in position 6 of hexoses have different chemical shifts, which is not the case here.

How do you explain that there are four branching residues and only one terminal residue?

Figure 6

Control has a value of 1, so why %?

ll. 258-261

Revise end of sentence.

ll. 285, 370, 391, 408

Undefined abbreviations: FRAP, FBS, DAPI, MTT

l. 297

“were” instead of “was”

l. 391

“stain” instead of “stained”

ll. 574-610

Rearrange reference numbers.

l. 604

“reducing” instead of “reducint”

Comments on the Quality of English Language

Few errors (corrected in comments above)

Round 2

Reviewer 4 Report

Comments and Suggestions for Authors

Manuscript number: molecules-2511878 (revised)

The authors corrected minor issues with the manuscript, but the problem of the identity and purity of the polysaccharide remains (contamination with mannan from yeast extract?). NMR assignments cannot be based on literature references that have themselves been based on literature references (Response 10 ref. 4; no NMR in ref. 5). The NMR spectra provided are not sufficient to extract the chemical shift assignments in Table 1, which are still not convincing, even after changing a few numbers.

Round 3

Reviewer 4 Report

Comments and Suggestions for Authors

I have checked the modifications to this manuscript and found no more issues with the NMR interpretation. However, the authors cannot claim that they determined the structure of the polysaccharide (ll. 67-69), but only a few structural characteristics, as mentioned elsewhere in the title and the text.
